# MMLU-Pro+: Evaluating Higher-Order Reasoning and Shortcut Learning in LLMs

## Abstract

Existing benchmarks for large language models (LLMs) increasingly struggle to differentiate between top-performing models, underscoring the need for more challenging evaluation frameworks. We introduce MMLU-Pro+, an enhanced benchmark building upon MMLU-Pro to assess shortcut learning and higher-order reasoning in LLMs. By incorporating questions with multiple correct answers across diverse domains, MMLU-Pro+ tests LLMs' ability to engage in complex reasoning and resist simplistic problem-solving strategies. Our results show that MMLU-Pro+ maintains MMLU-Pro's difficulty while providing a more rigorous test of model discrimination, particularly in multi-correct answer scenarios. We introduce novel metrics like shortcut selection ratio and correct pair identification ratio, offering deeper insights into model behavior and anchoring bias. Evaluations of six state-of-the-art LLMs reveal significant performance gaps, highlighting variations in reasoning abilities and bias susceptibility. We release the dataset and evaluation codes at `https://github.com/....`

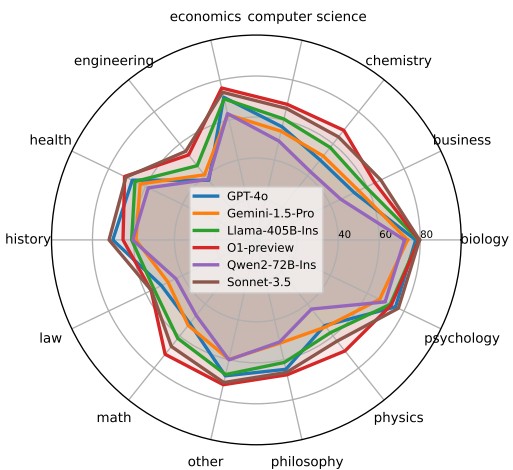

Figure 1: LLMs performance on our new MMLU-Pro+.

## 1 Introduction

Recent advancements in large language models (LLMs) have led to remarkable improvements in various natural language processing tasks [7, 15]. State-of-the-art models such as GPT-4o, O1-

Submitted to 38th Conference on Neural Information Processing Systems (NeurIPS 2024). Do not distribute.

preview, [1], Claude-3.5 Sonnet [2], Gemini [17], and Llama [20] have demonstrated impressive capabilities across a wide range of applications. However, as these models continue to evolve, existing benchmarks are struggling to keep pace, often reaching performance saturation and failing to effectively differentiate between model capabilities [10].

LLMs have been evaluated using a variety of benchmark datasets designed to test different aspects of language understanding and generation. Some of the most prominent benchmarks include IFEval [26] for instruction following, BBH (Big-Bench Hard) [19] for challenging reasoning tasks, MATH [25] for mathematical problem-solving, GPQA [13] for general-purpose question answering, and MUSR [27] for multi-task language understanding. These join other established benchmarks like the General Language Understanding Evaluation (GLUE) [22] and its successor SuperGLUE [21], as well as the Stanford Question Answering Dataset (SQuAD) [16] for reading comprehension.

Among these benchmarks, the Massive Multitask Language Understanding (MMLU) benchmark [12] has been widely adopted as a standard for evaluating LLMs due to its broad coverage of subjects. However, recent studies have shown that top-performing models are achieving near-identical scores on MMLU, with several models scoring between 86-87% accuracy [8]. This saturation raises concerns about the benchmark's ability to measure future advancements in LLM capabilities [18]. In response to these limitations, researchers developed MMLU-Pro [23], an iteration of the original MMLU designed to challenge LLMs with more complex, reasoning-focused questions and a greater number of answer options. While MMLU-Pro made significant strides, we identified further areas for enhancement that could improve the benchmark's ability to evaluate LLMs more effectively.

Recent research has highlighted the challenges of shortcut learning, where models exploit superficial patterns rather than developing deeper understanding[3], and the importance of evaluating higher-order reasoning in language models. Geirhos et al. [11] provide a comprehensive overview of shortcut learning in deep neural networks, emphasizing its prevalence and impact on model performance. Wei et al. [24] demonstrate how chain-of-thought prompting can elicit more sophisticated reasoning, addressing some of the limitations that lead to shortcut learning. The need for more nuanced evaluation methods has been underscored by Bowman and Dahl [6], who argue for fixing benchmarking in natural language understanding. Higher-order reasoning involves complex cognitive processes such as analysis, synthesis, and evaluation of multiple pieces of information [4]. It requires models to go beyond simple pattern matching or recall, engaging in more sophisticated thought processes that mimic human-like reasoning more closely. [14].

In this paper, we introduce MMLU-Pro+, a new benchmark that builds upon MMLU-Pro by incorporating these insights. The novelty of MMLU-Pro+ lies in its fundamental change to the nature of reasoning required from LLMs. By introducing questions with multiple correct answers, MMLU-Pro+ requires models to: a) Evaluate the validity of multiple statements independently; b) Recognize the potential for more than one correct answer; c) Discern subtle differences between correct and incorrect information; d) Resist the tendency to anchor on a single answer.

This approach increases the complexity of the benchmark, forcing models to engage in higher-order reasoning, recognizing and evaluating nuanced or multi-faceted concepts rather than relying on memorized patterns or simplistic heuristics. It tests a model's robustness and allows for better discrimination between models with varying levels of understanding and reasoning capabilities.

MMLU-Pro+ contributes to the field in several ways:

1. It specifically targets the reduction of shortcut learning by introducing questions with multiple correct answers.
2. It provides a more realistic evaluation scenario that mirrors real-world complexity, where problems often have multiple valid solutions.
3. It introduces new metrics such as the shortcut selection ratio and correct pair identification ratio, offering a more nuanced understanding of model performance beyond simple accuracy.

By addressing these aspects, MMLU-Pro+ serves as a reliable and informative tool for tracking progress in language understanding. It contributes to the ongoing efforts, highlighted by Bommasani et al. [5], to better understand and evaluate the capabilities and limitations of foundation models in AI, specifically targeting the reduction of shortcut learning and the promotion of higher-order reasoning skills in LLMs.

## 2  Dataset Construction

The construction of MMLU-Pro+ involves a systematic and scalable approach to modifying the original MMLU-Pro dataset, introducing multiple correct answers and various types of distractors to enhance its ability to evaluate higher-order reasoning skills in LLMs.

We begin with the MMLU-Pro dataset [23], which encompasses questions from 14 diverse domains including mathematics, physics, chemistry, law, engineering, psychology, and health. The initial dataset contains over 12,000 questions, each with up to ten answer options.

### 2.1  Dataset Modification Process

We modify the MMLU-Pro dataset in three distinct categories:

**True Positive Pairs.** For these questions, we introduce a "Both $X$ and $Y$ are correct" option, where $X$ is the original correct answer from MMLU-Pro, and $Y$ is a new correct option generated using GPT-4o. The process for generating $Y$ varies depending on the question type: a) For mathematical questions, we prompt the LLM to rewrite numbers or equations in alternative formats. b) For other types of questions, we instruct the LLM to find another correct option not already mentioned in the original choices, or to present the same correct information in a different, more complex way beyond simple paraphrasing. This process can be represented as:

$$Q_{TPP} = f_{LLM}(Q_{original}, \text{LLM}) \tag{1}$$

where $Q_{TPP}$ is the modified question with True Positive Pairs, $f_{LLM}$ is the LLM-based modification function, and LLM refers to GPT-4o.

**Partial False Positive Pairs.** For these questions, we create a "Both $X$ and $Y$ are correct" option where $X$ is the original correct answer from MMLU-Pro, and $Y$ is a randomly selected incorrect option from the original set of options.

**Complete False Positive Pairs.** For these questions, we create a "Both $X$ and $Y$ are correct" option where $X$ and $Y$ are two randomly selected incorrect options from the original set of options.

This composition allows for a comprehensive evaluation that tests more complex multi-answer reasoning skills, while also challenging LLMs to identify partially or completely incorrect option pairs. From the total of 12,032 questions in the dataset, 3,718 were modified using an LLM to create True Positive Pairs, while 4,153 were modified without LLM intervention. Specifically, 2,029 questions were modified with two wrong options (Complete False Positive Pairs), and 2,124 were modified with one correct and one wrong option (Partial False Positive Pairs). This ensures a robust evaluation across various question types and modification strategies. Incorporating True Positive Pairs tests the models ability to recognize multiple correct answers, reflecting real-world scenarios where different solutions can be equally valid. Meanwhile, the Partial and Complete False Positive Pairs test the models' ability to discern subtle inaccuracies and resist the tendency to assume correctness when presented with familiar information. This approach not only assesses a model's knowledge but also its capacity for nuanced reasoning and its robustness against potential shortcuts or biases in answering multiple-choice questions.

### 2.2  Post-Processing and Quality Assurance

To ensure the integrity of MMLU-Pro+, we implemented a rigorous post-processing and validation protocol:

*Human Auditing.* We conducted a comprehensive audit of 100 samples from each group (True Positive, Partial False Positive, and Complete False Positive Pairs) verifying the accuracy and appropriateness of new options.

*Consistency Checks.* We performed thorough checks across the entire dataset to ensure newly added options maintain the same style as original ones, preventing unintended evaluation biases.

*Error Identification.* We systematically identified and flagged potential inconsistencies or errors introduced during the modification process.

117 *Task Differentiation.* We ensured that the process of creating true positive pairs differs fundamentally
118 from the task of answering questions, minimizing the risk of model-specific advantages.

119 *Comprehensive Metrics.* Our evaluation metrics assess not only accuracy but also bias and shortcut
120 learning across diverse models, providing a holistic view of model performance.

121 While GPT-4o was used for creating True Positive pairs, our evaluation process is designed to be
122 model-agnostic. This approach ensures that the augmentation process genuinely increases question
123 difficulty rather than introducing biases favoring any particular model. These measures collectively
124 maintain MMLU-Pro+'s ability to differentiate between model capabilities, regardless of which LLM
125 was involved in the augmentation process. This is further validated in our experiments, where GPT-4o
126 is not the top-performing model, demonstrating the benchmark's independence from its creation
127 process.

## 3 Experiments

129 We evaluated several state-of-the-art LLMs, including two open-source models (LLaMA-3.1-405B-
130 Instruct [20] and Qwen-2-72B-Instruct [9]) and four closed-source models (OpenAI's GPT-4o [1] and
131 O1-preview, Claude-Sonnet-3.5 [2], and Gemini-1.5-Pro [17]), using the MMLU-Pro+ benchmark.
132 Models were assessed not only on accuracy but also on their ability to recognize and correctly select
133 the "Both X and Y are correct" option and avoid partially correct options, thereby demonstrating
134 higher-order reasoning and shortcut leanring.

### 3.1 Accuracy Analysis on MMLU-Pro+

136 Figure 1 illustrates the overall performance of each model on the MMLU-Pro+ dataset, while Table 1
137 provides a detailed breakdown of performance across individual subject categories, including the
138 comparative drop from MMLU-Pro. These results offer insights into the models' capabilities and the
139 increased challenge presented by MMLU-Pro+.

140 O1-preview demonstrates superior performance across the majority of categories, achieving the
141 highest average accuracy. The consistent outperformance suggests a more robust capability in
142 handling the increased complexity introduced by MMLU-Pro+. The universal decrease in accuracy
143 from MMLU-Pro to MMLU-Pro+ validates the increased challenge of our new benchmark. Notably,
144 O1-preview exhibits the smallest average performance drop, indicating enhanced resilience to the
145 modified question format and the introduction of multiple correct answers.

146 The engineering category reveals an interesting divergence, with Gemini-1.5-Pro and Sonnet-3.5
147 showing markedly smaller performance drops compared to other models. Conversely, the law
148 category presents an anomaly, with minimal performance drops and even slight improvements for
149 some models, suggesting that the MMLU-Pro+ modifications may have had less impact on this
150 domain. GPT-4o exhibits the largest average performance drop, indicating a potential sensitivity to
151 the structural changes in MMLU-Pro+. This suggests that high performance on standard benchmarks
152 may not necessarily translate to robust reasoning capabilities in more complex scenarios.

153 The performance gap between different models on MMLU-Pro, particularly the superior performance
154 of models like O1-preview and Claude-3.5-Sonnet compared to GPT-4o (which was used in dataset
155 creation), further validates our dataset construction methodology. Despite GPT-4o's involvement
156 in generating True Positive pairs, it does not exhibit an advantage in the question-answering task.
157 This discrepancy in performance across models suggests that MMLU-Pro successfully challenges
158 the LLMs, requiring genuine reasoning capabilities rather than pattern matching or exploitation of
159 dataset artifacts.

160 We also measured models' performance on (1) questions with two correct options (`both_correct`),
161 (2) questions with one correct and one incorrect option (`correct_and_wrong`), and (3) questions
162 with two incorrect options (`two_wrong`). The results, as illustrated in Figure 2, reveal significant
163 variations in model performance across these question types. Interestingly, all models showed lower
164 accuracy in the `both_correct` category compared to the other two, suggesting a potential difficulty
165 in identifying multiple correct answers. GPT-4o and Gemini-1.5-Pro showed similar performance
166 patterns, with their highest accuracies in the `correct_and_wrong` category. These findings highlight
167 the varying capabilities of different models in handling nuanced multiple-choice questions

Table 1: Accuracy (%) on MMLU-Pro+ Categories with Performance Drop from MMLU-Pro

| Category | Qwen2-72B-Ins | Gemini-1.5-Pro | GPT-4o | Llama-405B-Ins | Sonnet-3.5 | O1-preview |
|---|---|---|---|---|---|---|
| biology | $72.3_{-9.7}$ | $73.8_{-9.9}$ | $77.8_{-10.4}$ | $79.2_{-6.0}$ | $\mathbf{79.6}_{-7.8}$ | $79.3_{-9.9}$ |
| business | $45.8_{-22.2}$ | $55.5_{-16.6}$ | $53.1_{-26.5}$ | $59.4_{-17.6}$ | $\mathbf{67.4}_{-12.2}$ | $64.3_{-23.7}$ |
| chemistry | $42.5_{-16.3}$ | $52.4_{-17.7}$ | $49.9_{-25.7}$ | $57.8_{-15.1}$ | $64.3_{-12.3}$ | $\mathbf{68.5}_{-17.2}$ |
| computer science | $49.5_{-18.5}$ | $54.4_{-14.5}$ | $56.6_{-23.6}$ | $60.5_{-13.9}$ | $65.9_{-14.9}$ | $\mathbf{67.9}_{-67.9}$ |
| economics | $63.3_{-13.3}$ | $62.8_{-13.9}$ | $71.4_{-11.1}$ | $70.6_{-9.8}$ | $73.9_{-8.5}$ | $\mathbf{76.1}_{-9.6}$ |
| engineering | $37.8_{-9.8}$ | $40.7_{-3.9}$ | $37.3_{-18.5}$ | $46.3_{-13.5}$ | $\mathbf{55.3}_{-4.2}$ | $53.0_{-15.6}$ |
| health | $58.6_{-8.4}$ | $63.0_{-4.4}$ | $67.2_{-8.1}$ | $65.8_{-6.5}$ | $70.7_{-6.1}$ | $\mathbf{71.4}_{-7.7}$ |
| history | $60.4_{-6.0}$ | $58.8_{-7.3}$ | $70.1_{-2.4}$ | $60.9_{-6.6}$ | $\mathbf{71.9}_{-1.3}$ | $65.1_{-9.4}$ |
| law | $43.6_{-0.7}$ | $47.8_{-0.5}$ | $51.4_{-3.3}$ | $55.4_{-1.2}$ | $56.9_{-7.3}$ | $\mathbf{57.0}_{-11.5}$ |
| math | $47.2_{-23.4}$ | $53.4_{-8.8}$ | $52.1_{-25.9}$ | $61.5_{-15.7}$ | $66.5_{-9.8}$ | $\mathbf{71.4}_{-18.8}$ |
| other | $60.1_{-6.0}$ | $59.8_{-10.3}$ | $68.1_{-9.8}$ | $67.4_{-5.7}$ | $71.4_{-6.7}$ | $\mathbf{72.5}_{-8.5}$ |
| philosophy | $51.1_{-8.2}$ | $51.7_{-11.3}$ | $64.8_{-6.8}$ | $61.4_{-4.8}$ | $66.3_{-8.2}$ | $\mathbf{67.5}_{-12.0}$ |
| physics | $43.2_{-18.2}$ | $54.3_{-14.9}$ | $53.5_{-21.6}$ | $58.0_{-14.2}$ | $63.2_{-13.4}$ | $\mathbf{69.4}_{-17.7}$ |
| psychology | $69.8_{-6.4}$ | $66.8_{-9.7}$ | $75.4_{-5.9}$ | $72.5_{-4.8}$ | $\mathbf{76.9}_{-5.6}$ | $73.1_{-11.8}$ |
| Average | $53.2_{-11.9}$ | $56.8_{-10.2}$ | $60.6_{-14.3}$ | $62.6_{-9.5}$ | $67.9_{-8.5}$ | $\mathbf{68.3}_{-7.5}$ |

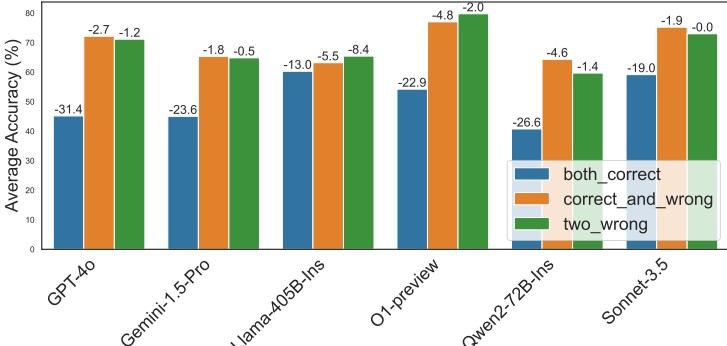

Figure 2: Accuracy on the three modified groups of questions. The amount of drop w.r.t original MMLU-Pro is written on the bars.

### 3.2 Analysis of Anchoring Bias and Shortcut Learning in MMLU-Pro+

Figure 3 illustrates the propensity of various language models to maintain their original choices when presented with modified questions in MMLU-Pro+, specifically for True Positive Pairs. To quantify this behavior, we introduce the Shortcut Selection Ratio (SSR), defined as follows:

$$\text{SSR}_{\text{wrong}} = \frac{N_{\text{stayed\_wrong}}}{N_{\text{total\_TPP}}} \quad (2)$$

$$\text{SSR}_{\text{partial}} = \frac{N_{\text{stayed\_partial}}}{N_{\text{total\_TPP}}} \quad (3)$$

Where $N_{\text{stayed\_wrong}}$ is the number of times the model stayed on a previously chosen wrong answer, $N_{\text{stayed\_partial}}$ is the number of times the model stayed only on the previously correct answer without acknowledging the newly introduced correct option, and $N_{\text{total\_TPP}}$ is the total number of True Positive Pairs.

This "shortcut selection ratio" provides insights into potential anchoring bias and shortcut learning behaviors. The graph reveals that all models exhibit a tendency to stick with their initial selections, both for previously wrong and partially correct options, suggesting a degree of anchoring bias. This behavior is particularly pronounced in GPT-4o and Qwen2-72B-Ins, which show higher rates of maintaining their original choices.

The persistence in selecting previously incorrect options (high $\text{SSR}_{\text{wrong}}$) is especially noteworthy, as it indicates potential limitations in these models' ability to reassess and engage in higher-order

reasoning when presented with new, valid alternatives. Similarly, a high $\text{SSR}_{\text{partial}}$ suggests a failure to recognize newly introduced correct options. Conversely, Gemini 1.5 Pro, Sonnet-3.5, and Llama 1.3 405B demonstrate lower shortcut selection ratios, suggesting a greater capacity for adapting their reasoning in light of new information. Interestingly, O1-preview has improved compared to GPT-4o, but still lags behind Gemini-1.5-Pro and Sonnet-3.5. These findings highlight the challenges language models face in fully leveraging the additional correct options introduced in MMLU-Pro+, and underscore the importance of developing benchmarks that can effectively evaluate and promote higher-order reasoning.

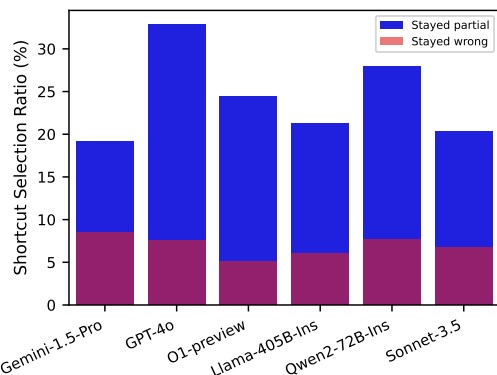

Figure 3: Shortcut Selection Ratio for True Positive Pairs in MMLU-Pro+

## 3.3 Analysis of Correct Pair Identification in MMLU-Pro+

In this experiment, we evaluate the models' ability to accurately identify true positive pairs among various types of answer combinations in MMLU-Pro+, providing insights into their reasoning capabilities and resilience to distractors.

Figure 4 presents an error analysis for various models on MMLU-Pro+, introducing a metric called the correct pair identification ratio. This ratio is defined as:

$$\text{Correct Pair Identification (CPI) Ratio} = \frac{N_{\text{TPP}}}{N_{\text{PFPP}} + N_{\text{CFPP}}} \qquad (4)$$

where $N_{\text{TPP}}$ represents the number of correctly identified True Positive Pairs, $N_{\text{PFPP}}$ is the number of times the model incorrectly predicted the Partial False Positive Pair, and $N_{\text{CFPP}}$ is the number of times the model incorrectly predicted the Complete False Positive Pair. This ratio measures the model's ability to identify correct pairs relative to its tendency to be misled by partially or completely incorrect pairs.

Sonnet-3.5 achieves the highest ratio (10.26), demonstrating superior discrimination capability in distinguishing correct answer pairs from misleading options. This suggests enhanced resistance to distractors and a more robust grasp of subject matter and question structure. The significant variation in ratios, ranging from 2.80 (Llama-405B-Ins) to 10.26 (Sonnet-3.5), reveals substantial differences in models' higher-order reasoning capabilities. Models with higher ratios exhibit a greater capacity for nuanced understanding, correctly identifying multiple true statements while rejecting plausible but incorrect combinations. These findings highlight MMLU-Pro+'s effectiveness in differentiating models based on their ability to handle complex, multi-correct answer scenarios, underscoring the importance of sophisticated evaluation metrics in assessing advanced language models. Some samples of the True Positive Pairs from the MMLU-Pro+ dataset can be seen in Figure 5.

## 4 Interpretation and Significance of Novel Metrics

The Shortcut Selection Ratio (SSR) and Correct Pair Identification (CPI) Ratio provide insights into model behavior that are directly relevant to real-world applications:

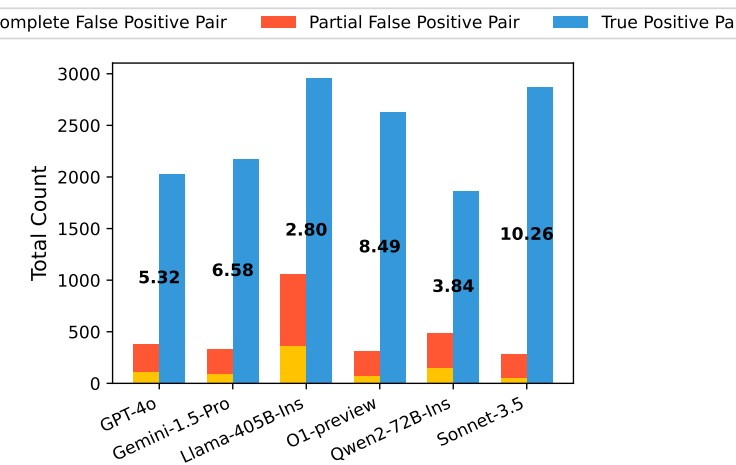

Figure 4: Error Analysis: Correct Pair Identification (CPI) in MMLU-Pro+. The numbers on the bars represent the CPI ratio values. A higher CPI ratio indicates better performance in distinguishing correct answer pairs from incorrect ones.

```
Question (Math): What number multiplied by 4 equals 36?

Options:
[A] 11, [B] 10, [C] 12, [D] 6, [E] 15, [F] 9, [G] 7, [H] 8, [I] 13, [J] 14, [K] 36/4, [L] Both 9 and 36/4 are
correct.

Predictions: Sonnet-3.5: L, O1-preview: L, Gemini-1.5-Pro: F, GPT-4o: F, Llama-405B-Ins: F, Qwen2-72B-Ins: F
```

```
Question (Computer Science): The disadvantage of Grid search is

Options:
[A] It cannot handle categorical variables., [B] It cannot be applied to non-differentiable functions., [C]
It is not suitable for datasets with a large number of features., [D] It cannot handle missing data., [E] It
cannot be used with neural networks., [F] It runs reasonably slow for multiple linear regression., [G] It can
only be used for binary classification problems., [H] It is hard to implement., [I] It cannot be applied to
non-continuous functions., [J] It is computationally expensive for large datasets., [K] Both It runs
reasonably slow for multiple linear regression and It is computationally expensive for large datasets are
correct.

Predictions: Sonnet-3.5: K, O1-preview: J, Gemini-1.5-Pro: J, GPT-4o: J, Llama-405B-Ins: J, Qwen2-72B-Ins: J
```

Figure 5: True Positive Pair Samples from `math` and `computer science` categories with model predictions.

SSR: A low SSR indicates a model's ability to adapt its reasoning when presented with new, valid information. This is crucial in dynamic decision-making environments, such as medical diagnosis or financial analysis, where new data may necessitate re-evaluation of initial conclusions. CPI Ratio: A high CPI Ratio suggests a model's proficiency in distinguishing between subtly different correct and incorrect information combinations. This skill is essential in fields like legal analysis, scientific research, or policy-making, where the ability to discern fine-grained differences in complex information is paramount.

These metrics go beyond simple accuracy, offering a more nuanced understanding of a model's reasoning processes and its potential performance in complex, real-world tasks.

## 5   Discussion

In this paper, we introduced MMLU-Pro+, an enhanced benchmark designed to evaluate the higher-order reasoning capabilities of large language models. By incorporating questions with multiple correct answers and various types of distractors, MMLU-Pro+ provides a more challenging and discriminative evaluation framework than its predecessors.

Our experimental results demonstrate the effectiveness of MMLU-Pro+ in several key areas:

*Increased Difficulty.* All evaluated models showed a consistent drop in performance when moving from MMLU-Pro to MMLU-Pro+, confirming the increased challenge of our benchmark.

*Model Differentiation.* MMLU-Pro+ revealed substantial differences in model performance. While O1-preview outperformed other models across most categories in general assessments, its dominance diminishes when anchoring bias and higher-order reasoning are considered.

*Anchoring Bias and Shortcut Learning.* The shortcut selection ratio analysis exposed varying degrees of anchoring bias across models, highlighting the challenge LLMs face in adapting their reasoning when presented with new, valid alternatives. Notably, Gemini-1.5-Pro and Sonnet-3.5 demonstrated less reliance on shortcuts.

*Higher-Order Reasoning.* The correct pair identification ratio provided insights into models' abilities to distinguish genuinely correct answer pairs from misleading options, with significant variations observed across different models. Sonnet-3.5 significantly outperformed others in this aspect.

These findings underscore the importance of new evaluation metrics in assessing advanced language models, particularly in scenarios requiring discernment between subtly different correct and incorrect information combinations.

MMLU-Pro+ not only serves as a more reliable and informative benchmark for tracking progress in LLM evaluation but also highlights areas for improvement in current models. The observed anchoring bias and varying abilities to identify correct pairs suggest that even top-performing models may still rely on simplistic heuristics or struggle with truly nuanced reasoning in complex scenarios.

Future work could explore the development of training techniques that specifically target the higher-order reasoning skills evaluated by MMLU-Pro+. Additionally, extending this benchmark approach to other domains or task types could provide a more comprehensive evaluation of LLM capabilities across diverse applications. While we used GPT-4o in our dataset construction process, our results demonstrate that this does not confer an unfair advantage to these or similar models. The significant performance drop of GPT-4o on the modified data, coupled with the superior performance of other models like Claude-3.5-Sonnet, indicates that our methodology produces a dataset that genuinely challenges LLMs. However, future work could explore alternative methods for dataset augmentation to further mitigate any potential biases introduced by LLM-assisted generation.

By providing a more challenging and discriminative benchmark, MMLU-Pro+ contributes to the ongoing effort to develop more capable and robust language models with human-like reasoning and understanding.

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
