# OpenReview forum: "MMLU-Pro+: Evaluating Higher-Order Reasoning and Shortcut Learning in LLMs"
_NeurIPS.cc/2024/Workshop/SafeGenAi — SafeGenAi Poster_

### Official Review · Reviewer_RxQj · 2024-10-09
**Review for MMLU-Pro+ Paper**

**Rating:** 7
**Confidence:** 4

**Review:**

# Summary
The paper proposes a significant modification of the existing MMLU-Pro dataset and evaluates various language models on the various subjects. This modification specifically provides LLMs with false information so as to distract the LLM. The degradation in performance is significant across various LLMs.

# Comments
- The paper makes a contribution to the community by making an automatic augmentation to an existing dataset, rather than hand-crafting the dataset. The dataset space can be quite large in this approach, as well, which is great.
- The paper can focus in the future on finetuning with this new dataset in the future.
- There is a clear alignment with the workshop's call for papers, as confusing agents to produce false or misleading information can be unsafe in various scenarios. However, in the future, the authors should focus unsafe responses generated using misleading prompts.

---

### Official Review · Reviewer_7Dxg · 2024-10-09
**Evaluation of  MMLU-Pro+: Evaluating Higher-Order Reasoning and Shortcut Learning in LLMs**

**Rating:** 7
**Confidence:** 3

**Review:**

The article introduces MMLU-Pro+, a novel benchmark for assessing higher-order reasoning and shortcut learning in large language models (LLMs). The experimental design is robust, leveraging multiple cutting-edge LLMs to demonstrate MMLU-Pro+'s ability to discern variations in model performance. While the results are promising and underscore the academic and practical significance of this benchmark, the manuscript would benefit from additional details regarding the experimental setup. Specifically, the authors should provide comprehensive information about model configurations and hyperparameters to enhance transparency and facilitate the reproducibility of the study.
To further substantiate the practical implications of the MMLU-Pro+ benchmark, it is recommended that the authors elaborate on the newly proposed evaluation metrics. A thorough discussion on how these metrics can be utilized in real-world applications would be valuable, highlighting their relevance beyond the context of this research. This refinement would solidify the benchmark's position as a pivotal tool for driving advancements in LLM evaluation and development.

---

### Official Review · Reviewer_25Lq · 2024-10-09
**MMLU-Pro+**

**Rating:** 7
**Confidence:** 4

**Review:**

The paper presents MMLU-Pro+, an enhanced benchmark for large language models (LLMs) that challenges the models' reasoning capabilities. The main advantages of previous work, such as MMLU, are as follows:

1. Motivation: The authors underscore the difficulty in differentiating between top-performing LLMs with existing benchmarks. They emphasize the necessity for more rigorous evaluation methods to test higher-order reasoning and reduce shortcut learning, where models exploit superficial patterns.

2. Building on MMLU-Pro, MMLU-Pro+ incorporates questions with multiple correct answers to test models' abilities to evaluate the validity of multiple statements, recognize multiple possible answers, discern subtle differences, and avoid anchoring on a single response. This approach compels LLMs to engage in more complex reasoning processes.

3. The paper introduces new metrics (the shortcut selection ratio and correct pair identification ratio), providing deeper insights into the models' behavior.

4. Evaluations of six state-of-the-art LLMs using MMLU-Pro+ reveal significant performance gaps, indicating variations in reasoning abilities and susceptibility to bias among these models.